# Using Fitbit data to examine factors that affect daily activity levels of college students

**Cheng Wang** [1] *, **Omar Lizardo** [2], **David S. Hachen** [3]

**1** Department of Sociology, Wayne State University, Detroit, MI, United States of America, **2** Department of Sociology, University of California Los Angeles, Los Angeles, CA, United States of America, **3** Department of Sociology, University of Notre Dame, Notre Dame, IN, United States of America

* chengwang@wayne.edu

**Data Availability Statement:** All relevant data in this study come from the NetHealth project, which are publicly accessible via the following URL: http://sites.nd.edu/nethealth/data-2/.

## Abstract

To date, the effect of both fixed and time-varying individual, social, psychological, environmental, and behavioral characteristics on temporal growth trends in physical activity (PA) among younger individuals remains an under-studied topic. In this paper, we address this gap in previous work by examining how temporal growth trends in PA respond to changing social, environmental, and behavioral characteristics using a large sample of college students (N = 692) who participated in the NetHealth project at the University of Notre Dame and from which fine-grained longitudinal data on physical activity and social interaction were collected unobtrusively via the use of wearables for 637 days (August 16, 2015 to May 13, 2017). These data are augmented by periodic survey data on fixed sociodemographic and psychological variables. We estimate latent growth-curve models for daily activity status, steps, active minutes, and activity calories. We find evidence of both a generalized friendship paradox and a peer effect for PA, with the average PA level of study participants' contacts being on average larger than their own, and with this average level exerting a statistically significant effect on individual PA levels. Notably, there was limited evidence of temporal growth in PA across the 637 days of observation with null temporal effects for three out of the four PA indicators, except for daily steps taken. Finally, we find that social, psychological, and behavioral factors (e.g., large network size, high extroversion levels, and more courses taken) are systematically associated with higher PA levels in this sample. Overall, our findings highlight the importance of social, environmental, and behavioral factors (such as peer networks and daily sociability) in modulating the dynamics of PA levels among college students.

## Introduction

Physical activity (PA) is one of the most consequential behavioral factors at the individual level, having a range of pervasive and systematic effects on a variety of relevant outcomes throughout the life-course. For instance, previous work shows that high levels of PA are systematically associated with lower morbidity and mortality rates, with less physically active individuals being more likely to die at younger ages than those who are more active [1]. The protective effects of PA on overall health and life expectancy operate via a variety of

**Funding:** This work was supported by National Institutes of Health #1 R01 HL117757-01A1. The funders had no role in study design, data collection and analysis, decision to publish, or preparation of the manuscript.

**Competing interests:** The authors have declared that no competing interests exist.

mechanisms, but the most important is its role in disease prevention and general immune system functioning. For instance, it has been shown that PA reduces the risk of hypertension, osteoporosis, heart disease, diabetes, colon and breast cancers, and the accumulation of immunosuppressive myeloid-derived suppressor cells (MDSCs) that are related to metastatic cancers [1–5]. In addition to its effects on physical health, PA has systematic effects on mental health outcomes (which themselves are linked to physical health and morbidity). For instance, previous work shows that PA boosts mental health by improving confidence, locus of control, self-esteem and self-identity and reducing symptoms of anxiety and depression [6–9].

## Physical activity and the life-course

Given the general salutary effects of PA on individual health and well-being, it is important to understand how levels of PA change as individuals grow and age, as well as the factors that help promote higher levels of PA. Some research indicates that PA decreases systematically as individuals age, with older people leading more sedentary and less active lives than younger people [10–21]. This work also shows that this secular decline in PA may begin as early as adolescence or even perhaps in childhood. Accordingly, transitions across educational institutional boundaries are prime periods in an individual's life where PA patterns are either maintained or begin a downward trajectory. The transition to college in particular, is considered to be one such critical juncture. As such, many health specialists see it as an opportune setting to devise cost-efficient intervention programs designed to develop "good" PA habits, given the easy availability of exercise professionals and facilities on campus [22–26]. The basic idea is that if behavioral patterns conducive to higher and consistent levels of PA can be developed during the college years, then individuals will be able to retain these PA habits later in life, thus being able to reap attendant physical and health benefits [27–29].

## What factors lead to higher PA?

To facilitate the development of PA among younger people in general and students making the transition to college in particular, health and social scientists need to better understand the factors that can *systematically* affect PA, both in terms of its levels and overall temporal growth. Keating et al. [30] conducted a thorough review and summarize these factors into four categories: *social* (e.g., peer-influence effects and overall social connectivity), *personal* (e.g., individual characteristics), *psychological* (e.g., personality and mental status), and *environmental* (e.g., weather conditions and facilities access).

**Social factors.** Peers are salient socializing agents for youth and playing a particularly important and influential role in helping shape adolescents' evolving social worlds [31]. Existing research finds that PA levels are influenced by a young person's peers [19, 26, 32–34]. Another social factor, the size of people's personal social network, has also been found to be positively related to one's PA level [35]. Using a standard measure for peer effects [36], the average level of PA among a person's contacts, we expect college students with more active peers to be more active. In the same way, we expect individuals who are more socially active, as given by the number of others who they sustained daily contacts, to be more physically active than people who have a smaller set of social interaction partners.

**Personal factors.** Previous work points to the importance of a variety of factors measured at the level of the individual in modulating PA levels. This includes identification with and membership in a variety of sociodemographic groups (e.g., based on gender, race, religion and so forth), as well as individual physical characteristics strongly linked to PA such as the body mass index (BMI). Regarding the first set of personal factors, men are generally found to have higher PA levels than women [10, 13, 16, 17, 22, 33, 37]. Other studies suggest that African

American and Asian American students are less physically active than white and Hispanic students [38]. Religious affiliation, on the other hand, has not been found to influence PA levels [39]. We expect such relationships to be reproduced in our analysis. With regard to BMI and physical traits, the bulk of the research suggests that PA is negatively associated with adiposity (as would be expected). However, the exact relationship between PA and physical markers such as the body mass index (BMI) is unclear [40]. As such, one of the goals of this analysis is to determine the precise functional form of the BMI-PA linkage in a sample of college students.

**Psychological factors.**   These include both relatively fixed personality traits, and chronic conditions associated with mental health such as depressive mood and anxiety. With regard to the former, traits extraversion and conscientiousness have been found to be positively associated with higher levels of PA [41]. However, pervious work reveals mixed results concerning the relationship between depression and PA, with some studies suggesting that increased levels of depression are likely to lead to more sedentary and less active lifestyles [42], and more recent research confirming the existence of a PA to depression connection among college students [43].

**Environmental/Situational factors.**   Naturally, individual levels of PA are expected to be moderated by a host of more transient environmental and situational factors (varying weekly, daily and at faster time scales). For instance, due to the rhythms of the social and academic calendar, college students walk more steps on weekdays than on the weekend (i.e., Saturday and Sunday) [44]. Students who are more academically active (e.g., taking more classes) are also expected to be more physically active. In the same way, we expect people to be more active on days where weather conditions are more pleasant and conducive to outdoor activities, like the summer months. Additionally, bad weather (rain, snowfall) is expected to decrease PA, by increasing the tendency to stay indoors. Despite these natural expectations, the exact relationship between weather and PA, as reviewed in [45], is still unclear. The present study leverages metadata on daily weather patterns to establish more precisely empirical connection between daily weather conditions and multiple indicators of PA measured at a corresponding time scale.

## Contributions of the present work

Overall, while previous studies have greatly contributed to our understanding of PA among college students, there are manifest gaps in the literature. First, no longitudinal study has been conducted to study the changes in PA among college students as well as how they are associated with corresponding determinants [46]. Second, prior ecological models do not take into consideration behavioral factors, such as sleep duration and number of classes taken by a college student, both of which may impact PA levels and their changes over time [47, 48]. Third, while steps assessed by pedometers and calories measured by accelerometers are good measures of PA, there is no consensus in the literature about how to standardize PA levels so that they can be compared across studies [49, 50]. Finally, all the aforementioned factors have not been examined in a single modeling framework, a critical factor when the possibility of spurious effects results from multiple determinants. These gaps in literature are in large part the result of the paucity of longitudinal data tracing the PA of college students around the clock in natural settings over a long period of time.

The current study builds upon extant literature examining the importance of PA and factors related to PA among college students. We use high-quality, fine-grained data collected by Fitbit devices and smartphones between August 16, 2015 (i.e., the first day of orientation week during fall semester 2015) and May 13, 2017 (i.e., the last day of spring semester 2017) from

the NetHealth project, a longitudinal research of 692 undergraduate students at the University of Notre Dame [51–53]. While most prior research relies on self-reported and/or cross-sectional data, the alternative approach used in this study offers updated functionality and avenues of inquiry for researchers to systematically investigate the health trends of college students and accurately estimate how multiple categories of factors influence these trends.

## Materials and methods

### Data sources

The NetHealth project was supported by National Institute of Health (NIH) and approved by the Institutional Review Board at the University of Notre Dame. Written informed consents were obtained from all participants in the study. For participants who were minors at the time of study initiation informed consent was obtained from a parent or legal guardian.

In fall 2015 the University of Notre Dame admitted 2007 full-time freshmen, among which 1,069 (or 53%) were male students, 938 (or 47%) were female students, 1,352 (or 67%) were non-Hispanic white students, 219 (or 11%) were Hispanic students, 80 (or 4%) were non-Hispanic African-American students, 111 (or 6%) were non-Hispanic Asian-American students, 3 (or 0%) were non-Hispanic American Indian or Alaska Native students, 93 (or 5%) were non-Hispanic students with two or more races, 143 (or 7%) were students of other races, and 6 (or 0%) were students with race/ethnicity unknown (the detailed sampling frame is available from https://www3.nd.edu/~instres/CDS/2015-2016/CDS_2015-2016.pdf.) The NetHealth project team adopted a stratified sampling strategy and aimed for a specific percentage of each gender-race strata based on the sampling frame. The project team also made an estimation of the maximum number of Fitbit devices that could be distributed among the participants based on the NIH budget and sent out invitations to 730 students. Finally, 692 students accepted the invitations and completed the assent forms.

The NetHealth data are fully de-identified and publicly available via the *NetHealth Project* website (http://sites.nd.edu/nethealth/data-2/). We use five types of data from the NetHealth data in the following analyses:

1. Fitbit data collected from August 16, 2015 to May 13, 2017 containing information on 18 daily activity measures for each study participant, aggregated from their minute-by-minute movement and heart rate data (i.e., low range calories and minutes, fat burn calories and minutes, cardio calories and minutes, peak calories and minutes, steps, floors, sedentary minutes, lightly active minutes, fairly activity minutes, very active minutes, marginal calories, activity calories, calories BMR, and calories out; More detailed explanation of these items is available from the http://help.fitbit.com website) and sleeping minutes (using a proven algorithm based on the movement pattern and heart rate variability, or HRV) captured by the Fitbit Charge HR wristbands and stored at the Fitbit cloud, which was, in turn, backed up to a server administered by the NetHealth project team;

2. Smartphone data during the study period containing communication records and geolocation information for each study participant;

3. Weather data downloaded from the http://www.usclimatedata.com/ website, containing daily highest and lowest temperatures, precipitation (in inches), snowfall (in inches), and snow depth (in inches) in the geographic area in which the study was conducted;

4. Data from entry survey conducted during fall 2015 which contain each student's self-reported sex, race, religious preference, height and weight, psychological traits. These data are supplemented with four more waves of surveys conducted during winter 2016, summer

2016, winter 2017, and summer 2017, containing information on courses taken during fall semester 2015, spring semester 2016, fall semester 2016, and spring semester 2017 for each study participant; and

5. Course data containing the course names and schedules during fall semester 2015, spring semester 2016, fall semester 2016, and spring semester 2017.

These five types of data are merged to generate a daily physical activity, network, and weather data over 637 days for 619 persons.

## Measures

Students' daily PA status is generated as a standardized factor score of the 18 Fitbit items (Cronbach's $\alpha = 0.89$). We also include 3 ancillary dependent variables–daily steps, daily active minutes (i.e., the sum of very active minutes and fairly active minutes reported by Fitbit devices), and daily activity calories–as supplementary indicators of daily PA.

The main social network-based predictors of study participants' daily PA are *peer influence* measured by the average daily PA of the in-study contacts of each participant [36], and *daily network size* measured by the total number of contacts each study participant communicated with in a given day. The first measure indicates specific exposure to the PA-relevant behavior of daily contacts, while the latter measure represents general exposure to others via social interaction.

Additional predictors include participant's gender (0 = Male participant, 1 = Female participant), ethnic/racial identification (0 = White, 1 = Latino, 2 = African American, 3 = Asian American, 4 = Other races), religious preference (0 = Catholic, 1 = Protestant, 2 = Other religion, 3 = No religion), BMI (weight/height$^2$), extraversion (a standardized factor score of 8 items; Cronbach's $\alpha = 0.87$), agreeableness (a standardized factor score of 9 items; Cronbach's $\alpha = 0.80$), conscientiousness (a standardized factor score of 9 items; Cronbach's $\alpha = 0.83$), neuroticism (a standardized factor score of 8 items; Cronbach's $\alpha = 0.82$), openness (a standardized factor score of 10 items; Cronbach's $\alpha = 0.79$; the five standardized factor scores above often refer to as the big five factors in personality trait ratings [54]; see S1 Table for detailed items), depression (a standardized factor score of 20 items modified from the Center for Epidemiologic Studies Depression Scale; CES-D; Cronbach's $\alpha = 0.94$) [55], the highest and lowest temperatures (˚F), precipitation in inches, snowfall in inches, and snow piling depth in inches, sleeping minutes, and number of classes taken that day. In addition, we include the dummy variables for weekend/weekday comparison (0 = Sunday, 1 = Monday to Thursday, 2 = Friday or Saturday) and normal/specific day comparison (0 = Normal school day, 1 = Home football game Saturday, 2 = Midterm break, 3 = Winter break, 4 = Summer break, 5 = Thanksgiving holidays, 6 = Easter holidays, 7 = Orientation week, 8 = Final exam week). The ages of 99.6% participants ranged from 17 to 19 when they joined the project in 2015, with a mean of 18.4 and standard deviation of 1.8. Since the age variable is relatively homogenous among NetHealth participants, we do not include it in the model.

## Plan of analysis

First, we establish whether daily activity levels changed appreciably over the 637 days. To explore this time trends, we fit latent growth-curve models (LGCMs) with random/fixed effects to the daily panel data in Stata V15.2. Scripts for estimating the LGCMs are available at https://github.com/socnetfan/pa/. LGCMs are hierarchical linear models (HLMs) since they contain both within-subject and between-subject variations. The multi-level equations are shown below.

Level 1 within-subject equation:

$$y_{id} = \eta_{0i} + \eta_{1i}(\text{Day}) + \beta_{1d}x_{id} + \varepsilon_{id} \tag{1}$$

where $y_{id}$ is the dependent variable (DV) for the $i$th subject at the $d$th day, $\eta_{0i}$ is the latent factor indicating initial value of DV, $\eta_{1i}$ is the latent factor for linear slope of DV, $x_{id}$ is the time-varying covariate for the $i$th subject at the $d$th day, $\beta_{1d}$ is the estimated parameter for $x_{id}$, and $\varepsilon_{id}$ is the within-subject random error with $\varepsilon_{id} \sim N(0, \sigma_{id}^2)$.

Level 2 between-subject equations:

$$\eta_{0i} = \gamma_{00} + \gamma_{01}z_i + \delta_{0i} \tag{2}$$

$$\eta_{1i} = \gamma_{10} + \gamma_{11}z_i + \delta_{1i} \tag{3}$$

where $\gamma_{*0}$ are intercept terms, $z_i$ is the time-constant variable, $\gamma_{01}$ is the effect of time-constant variable on the intercept, $\gamma_{11}$ is the effect of time-constant variable on the linear slope, and $\delta_{*i}$ are between-subject random error terms.

Fig 1 illustrates the dependencies specified in the statistical model. The dependent variable $y$ measured at multiple waves with a random error $\varepsilon$ has three predictors: (1) the intercept or initial value $\eta_0$, (2) the linear slope or rate of change $\eta_1$, and (3) the time-varying variable $x$, reflecting the within-subject variation. Both $\eta_0$ and $\eta_1$ are estimated as latent variables. The effect of time-constant variable $z$ on the dependent variable $y$ goes through $\eta_0$ and $\eta_1$, indicating the between-subject variation. This method enables the time trend and effects of determinants to be analyzed in a unified (one-step) modeling framework.

## Sample attrition

All 692 NetHealth participants took the entry survey when they joined the project. The follow-up surveys conducted during winter 2016, summer 2016, winter 2017, and summer 2017 have a response rate of 91%, 83%, 80%, and 76%, respectively, resulting in a difference in the number of individuals we see in the latent growth-curve models. Following Faust et al. [51] and Wang et al. [53], we set 80% as a threshold for the compliant rate, i.e., the number of minutes that a Fitbit device is on the wrist divided by total minutes of a day (1440). In other words, we estimate the latent growth-curve models using data over the 637 days from participants who had relatively complete Fitbit data (defined as 80% or more daily within-person records) to guarantee the validity of PA measures (daily activity, steps, active minutes, and activity calories) and sleeping minutes. After applying the 80% threshold, the number of days of Fitbit data ranges from 1 to 278, with a mean of 148 and standard deviation of 75.

## Results

### Summary statistics

Table 1 shows the summary descriptive statistics (means and standard deviations) for all time-varying variables included in the analysis. Table 2 shows the corresponding descriptive for the time-constant (fixed) individual characteristics in the sample of study participants.

As shown in Table 1, an average NetHealth participant was less physically active than his or her in-study contacts, exhibiting lower daily PA status (0.01 vs. 0.03), walking fewer steps (11,258 vs. 11,648), spending fewer active minutes (44 vs. 47), and burning fewer activity calories (952 vs. 996) per day. This is consistent with the idea that the average level of any characteristic or trait of network contacts tends to exceed that of the focal individual, also known as the "generalized friendship paradox" [56–58]. In addition, on average, study participants

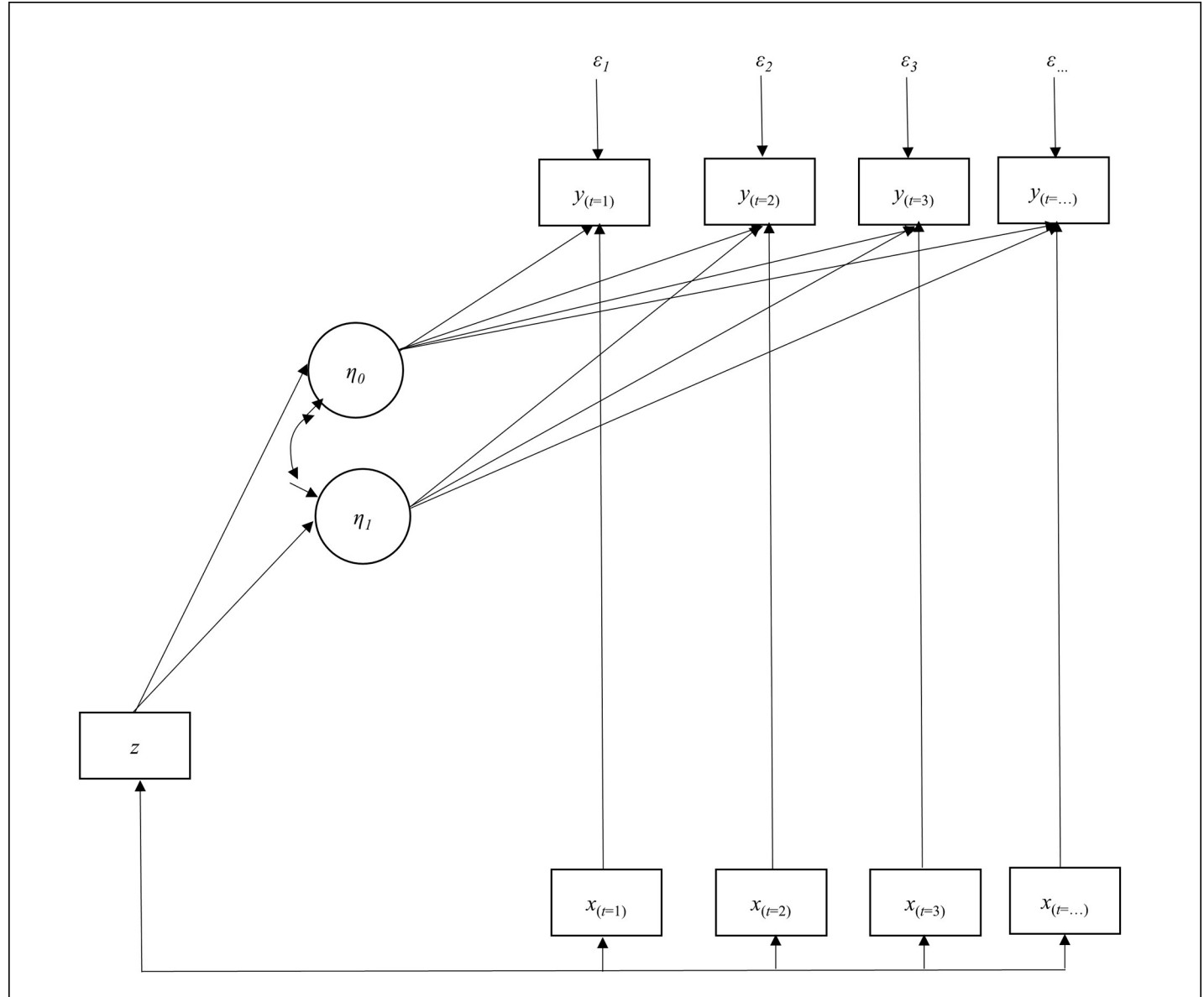

**Fig 1. Statistical framework of latent growth-curve model.** *y* represents the time-varying dependent variable such as daily activity, steps, active minutes, and activity calories; *x* represents the time-varying covariate among social, environmental, and behavioral factors that change on a daily basis; *z* represents the time-constant covariate among personal and cognitive factors that are stable over time; and $\eta_0$ and $\eta_1$ represent the latent variables for the intercept and linear slope of the dependent variable *y*, respectively.

communicated with 10 different contacts via smartphone, slept 370 minutes, took about 2.5 classes per day on normal school days, and spent about 74% time on campus between August 16, 2015 and May 13, 2017. The weather indicators during the 637 days are also shown.

Fig 2 shows a scatter plot and linear prediction plot with 95% confidence interval for each of the 4 dependent variables. While there were more outliers during fall semester 2015 and fall semester 2016 due to the home football games and PA levels were generally low during two winter breaks and one summer break, we do not see evidence of quadratic or piecewise linear trend. Therefore, latent growth-curve model is appropriate to examine the longitudinal pattern of PA levels over the 637 days.

**Table 1. Summary statistics for time-varying variables.**

| Individual-day variables | Mean (SD) or *n* (%) |
|---|---|
| Daily activity | 0.01 (0.58) |
| Daily steps | 11257.99 (5873.89) |
| Daily active minutes | 43.75 (57.24) |
| Daily activity calories | 952.10 (808.87) |
| In-study contacts' average daily activity | 0.03 (0.49) |
| In-study contacts' average daily steps | 11648.14 (5008.69) |
| In-study contacts' average daily active minutes | 46.65 (47.51) |
| In-study contacts' average daily activity calories | 996.41 (673.56) |
| Daily network size | 10.28 (7.33) |
| Daily in bed minutes | 370.06 (202.66) |
| Daily number of courses taken | 2.50 (1.17) |
| Daily on campus status (1 = yes) | 0.74 (0.10) |
| Number of cases | 269,057 (100.00%) |
| Day variables | |
| Weather indicators | |
| Highest temperature (˚F) | 58.39 (19.56) |
| Lowest temperature (˚F) | 40.22 (16.69) |
| Precipitation in inch | 0.12 (0.40) |
| Snowfall in inch | 0.17 (0.80) |
| Snow piling depth in inch | 0.44 (1.40) |
| Number of days | 637 (100.00%) |

Daily activity is a standardized factor score of 18 items, i.e., low range calories and minutes, fat burn calories and minutes, cardio calories and minutes, peak calories and minutes, steps, floors, sedentary minutes, lightly active minutes, fairly activity minutes, very active minutes, marginal calories, activity calories, calories BMR, and calories out.

## Latent growth-curve model results

Each of the four latent growth-curve models whose corresponding coefficient estimates are shown in Table 3 has a Root Mean Square Error of Approximation (RMSEA) fewer than .06 and Comparative Fit Index (CFI) greater than .95, both suggesting a good fit to the data. Looking at the overall growth trends, the models indicate that the NetHealth participants' daily PA status, active minutes, and activity calories did not appreciably change, upwards or downwards, between August 2015 and May 2017. We only detect a statistically significant downward trend for daily steps. Moreover, this declining pattern is mitigated among female participants who had lower initial levels of daily steps.

More importantly, we find a statistically significant peer influence: a unit increase in in-study contacts' average daily PA status yielded 0.06-unit increment in the focal student's daily PA status. Moreover, the interaction effect between participant's gender identification and PA indicates that female participants were less susceptible to peer influence than were male participants in all four models. For example, in Table 1 that in-study contacts had an average of 11,648 daily steps, which resulted in 831 steps increment for male participants but 606 steps increment for female participants, respectively. This is consistent with previous work showing that males of this age group tend to be more susceptible to the influence of other males in this social context [59], and in terms of PA in particular [34].

**Table 2. Summary statistics for time-constant variables.**

| Variables for each individual | Mean (SD) or *n* (%) |
|---|---:|
| Female participant (1 = yes) | 314 (50.81%) |
| Race/Ethnicity | |
| White (1 = yes) | 403 (65.32%) |
| Latino (1 = yes) | 80 (12.97%) |
| African American (1 = yes) | 37 (6.00%) |
| Asian American (1 = yes) | 57 (9.24%) |
| Other race (1 = yes) | 40 (6.48%) |
| Religious preference | |
| Catholic (1 = yes) | 453 (73.42%) |
| Protestant (1 = yes) | 66 (10.70%) |
| Other religion (1 = yes) | 26 (4.21%) |
| No religion (1 = yes) | 72 (11.67%) |
| Extraversion | -0.01 (0.72) |
| Agreeableness | 0.01 (0.60) |
| Conscientiousness | 0.04 (0.62) |
| Neuroticism | -0.02 (0.64) |
| Openness | -0.01 (0.59) |
| Depression scale score | 0.06 (0.45) |
| BMI | 22.82 (3.35) |
| Number of study participants | 619 (100.00%) |

Daily network size is also a statistically significant predictor of higher PA levels, even when time-constant personality characteristics, such as extroversion, and time-varying behavioral variables, such as the number of classes taken daily are adjusted for, both of which are reliable contributors of participants' daily activity. In terms of substantive significance, a unit increase in the number of daily contacts led to 18 more steps taken, 7 more seconds of active time, and 2 additional activity calories burned.

The effect of additional predictors is, for the most part, intuitive and in line with expectations. Female participants were less active (on average) than male participants, and Latino and African American students as well as students of other races were less active in comparison to students who identify as white. Higher BMI participants had higher daily PA status and activity calories, but walked fewer steps. People high on trait extraversion were more active than those low on this personality characteristic. Students were more physically active when on campus, as well as over Fridays and Saturdays, home football game Saturdays, warmer days, snowy days, days they got more sleep, and days on which they took more classes. Not surprisingly, participants were less active on Sundays, midterm and winter breaks, holidays spent at home, final exam weeks, and colder days.

As a robustness check, we estimate four sets of ancillary models. The first set of models includes the square term for the daily time trend and its interaction with other time-constant variables to check whether there were nonlinear growth trends across the four outcomes and whether the nonlinear component varied across fixed characteristics of study participants. We find no evidence of nonlinear time for any of the PA outcomes nor evidence of moderation of the nonlinear components by fixed individual characteristics.

The second set of ancillary models tests for the existence of nonlinear BMI effects by including a square term for this variable, but finds no evidence of nonlinearity in this predictor. The third set of models tests for nonlinear effects of daily highest temperature recorded, daily

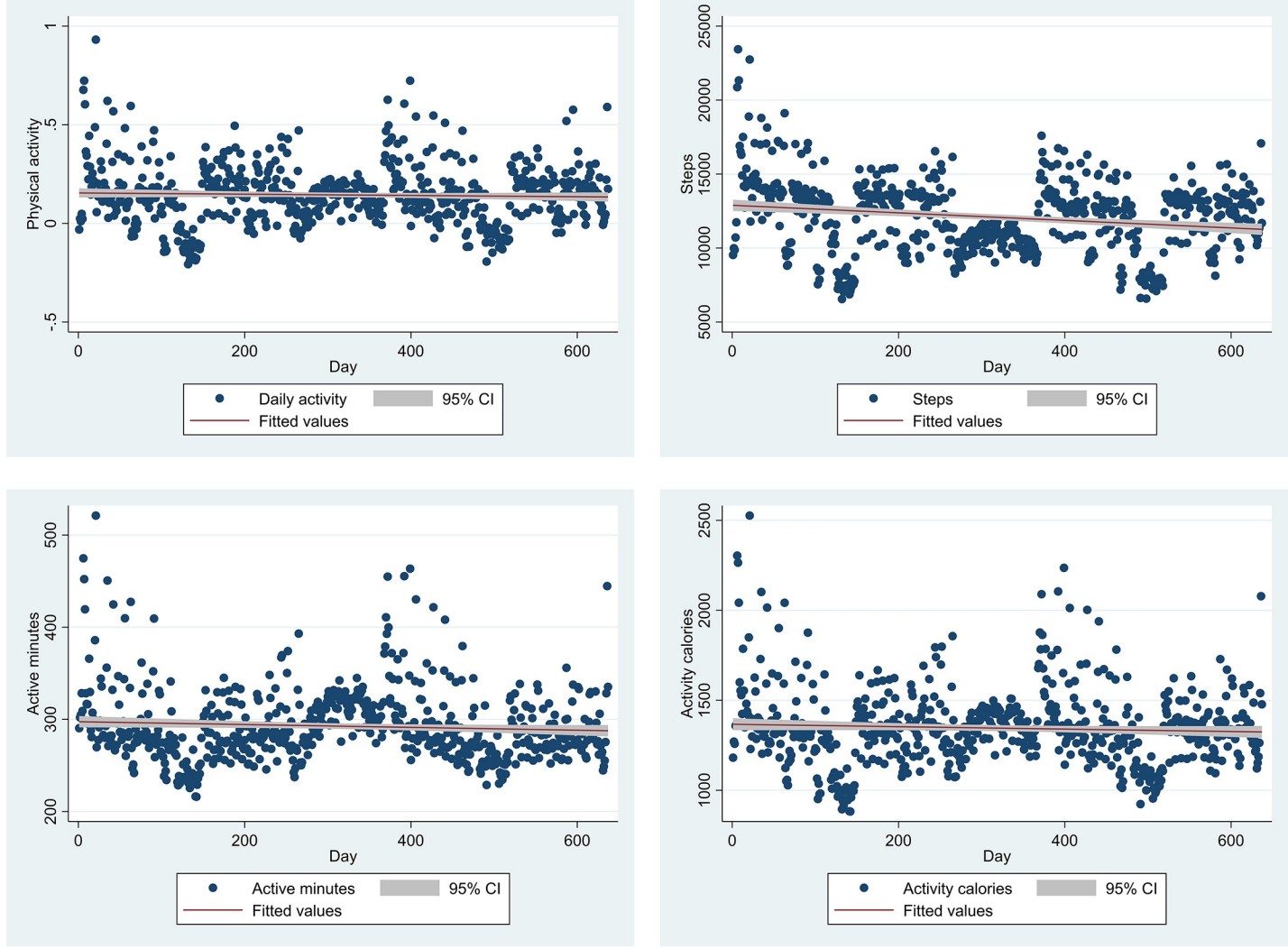

**Fig 2. Scatter plot and linear prediction plot with 95% confidence interval for each of the four dependent variables.** a. Physical activity; b. Steps; c. Active minutes; d. Activity calories.

lowest temperature records, and both nonlinear temperature effects in the same model. These curvilinear daily temperature effects are not statistically significant, nor do they help decrease the AIC and BIC values.

Finally, the fourth set of ancillary models adopts a backward elimination approach by taking out one control variable at a time from each of the model specifications shown in Table 3 till a downward time trend on PA is seen. Using this strategy, we find that when the participant's BMI and its effect on the linear slope are removed, the declining time trend increases from -8.28 to -4.63 for daily steps and becomes statistically significant for daily PA status and activity calories, though it remains statistically non-significant for daily active minutes. This suggests that omission of BMI-related variables would have resulted in an understatement of negative time trend for daily steps taken, and a spurious finding of a significant time trend for daily PA status and activity calories. Omitting BMI-related variables greatly increases the AIC and BIC values of each ancillary model in this set, suggesting these two variables must be adjusted for to obtain reliable and consistent parameter estimates.

**Table 3. Parameters and 95% confidence intervals from the latent growth-curve models.**

| | Daily activity | Steps | Active minutes | Activity calories |
|---|---|---|---|---|
| Average measure of in-study contacts | 0.06*** | 0.07*** | 0.07*** | 0.07*** |
| | (0.05, 0.07) | (0.06, 0.08) | (0.06, 0.08) | (0.06, 0.08) |
| Female participant × Average measure of in-study contacts | -0.05*** | -0.02* | -0.07*** | -0.07*** |
| | (-0.07, -0.04) | (-0.04, -0.00) | (-0.09, -0.05) | (-0.08, -0.05) |
| Daily network size | 0.00** | 17.52*** | 0.11* | 2.29*** |
| | (0.00, 0.00) | (9.35, 25.69) | (0.02, 0.20) | (1.33, 3.24) |
| Day | -0.00 | -8.28** | -0.04 | -0.58 |
| | (-0.00, 0.00) | (-14.43, -2.13) | (-0.12, 0.04) | (-1.39, 0.24) |
| Female participant (1 = yes) | -0.32*** | -756.50* | -18.09*** | -447.40*** |
| | (-0.38, -0.25) | (-1458.40, -54.67) | (-25.91, -10.27) | (-530.10, -364.80) |
| Female participant × Day | 0.00 | 3.59*** | 0.00 | 0.17 |
| | (-0.00, 0.00) | (1.83, 5.36) | (-0.02, 0.03) | (-0.05, 0.42) |
| Latino (1 = yes) | -0.13** | -341.20 | -10.57 | -159.00** |
| | (-0.22, -0.04) | (-1253.00, 570.50) | (-21.19, 0.04) | (-269.60, -48.45) |
| Latino × Day | 0.00 | 0.51 | 0.00 | 0.11 |
| | (-0.00, 0.00) | (-1.89, 2.92) | (-0.03, 0.04) | (-0.21, 0.43) |
| African American (1 = yes) | -0.14* | -679.80 | -19.01* | -147.00 |
| | (-0.27, -0.00) | (-2096.90, 737.20) | (-35.48, -2.54) | (-318.20, 24.15) |
| African American × Day | -0.00 | -2.28 | 0.00 | 0.01 |
| | (-0.00, 0.00) | (-5.87, 1.32) | (-0.05, 0.05) | (-0.47, 0.49) |
| Asians American (1 = yes) | -0.08 | 38.56 | -8.80 | -75.13 |
| | (-0.19, 0.02) | (-1094.40, 1171.50) | (-21.97, 4.37) | (-211.90, 61.61) |
| Asians American × Day | 0.00 | -0.09 | 0.00 | -0.02 |
| | (-0.00, 0.00) | (-3.00, 2.82) | (-0.04, 0.04) | (-0.40, 0.37) |
| Other races (1 = yes) | -0.21*** | -1337.30* | -20.37** | -285.30*** |
| | (-0.34, -0.09) | (-2646.80, -27.78) | (-35.65, -5.10) | (-444.50, -126.10) |
| Other races × Day | 0.00 | 0.44 | 0.00 | 0.12 |
| | (-0.00, 0.00) | (-2.93, 3.82) | (-0.04, 0.05) | (-0.33, 0.57) |
| Protestant (1 = yes) | 0.02 | 297.20 | -8.79 | -14.74 |
| | (-0.09, 0.12) | (-796.40, 1390.80) | (-21.54, 3.96) | (-147.10, 117.60) |
| Protestant × Day | 0.00 | 0.39 | 0.05* | 0.24 |
| | (-0.00, 0.00) | (-2.43, 3.21) | (0.01, 0.08) | (-0.13, 0.62) |
| Other religion (1 = yes) | -0.03 | -413.70 | -3.33 | -60.07 |
| | (-0.18, 0.12) | (-1996.60, 1169.30) | (-21.67, 15.00) | (-251.40, 131.30) |
| Other religion × Day | 0.00 | 2.96 | 0.02 | 0.35 |
| | (-0.00, 0.00) | (-1.00, 6.92) | (-0.03, 0.08) | (-0.17, 0.88) |
| No religion (1 = yes) | 0.01 | -62.23 | 2.19 | -30.44 |
| | (-0.09, 0.11) | (-1123.20, 998.70) | (-10.16, 14.54) | (-159.00, 98.14) |
| No religion × Day | 0.00 | -0.66 | -0.01 | 0.02 |
| | (-0.00, 0.00) | (-3.51, 2.19) | (-0.04, 0.03) | (-0.36, 0.40) |
| BMI | 0.01** | -147.70** | 0.50 | 20.37** |
| | (0.00, 0.02) | (-250.60, -44.79) | (-0.69, 1.69) | (7.96, 32.78) |
| BMI × Day | 0.00 | 0.16 | 0.00 | 0.01 |
| | (-0.00, 0.00) | (-0.11, 0.43) | (-0.00, 0.01) | (-0.02, 0.05) |
| Extraversion | 0.09*** | 1221.80*** | 6.98* | 149.80*** |
| | (0.05, 0.14) | (762.90, 1680.70) | (1.66, 12.31) | (94.37, 205.30) |

(*Continued*)

**Table 3.** (Continued)

|  | Daily activity | Steps | Active minutes | Activity calories |
|---|---|---|---|---|
| Extraversion × Day | -0.00 | -0.64 | -0.00 | -0.11 |
|  | (-0.00, 0.00) | (-1.82, 0.54) | (-0.02, 0.01) | (-0.27, 0.04) |
| Agreeableness | 0.02 | 296.00 | 4.02 | 43.13 |
|  | (-0.03, 0.08) | (-262.20, 854.20) | (-2.49, 10.53) | (-24.56, 110.80) |
| Agreeableness × Day | -0.00 | -0.17 | -0.01 | -0.10 |
|  | (-0.00, 0.00) | (-1.68, 1.33) | (-0.03, 0.01) | (-0.30, 0.10) |
| Conscientiousness | 0.02 | 364.80 | -0.95 | 0.59 |
|  | (-0.03, 0.07) | (-165.80, 895.50) | (-7.13, 5.23) | (-63.69, 64.87) |
| Conscientiousness × Day | 0.00 | 0.66 | 0.02 | 0.18 |
|  | (-0.00, 0.00) | (-0.75, 2.07) | (-0.00, 0.04) | (-0.01, 0.37) |
| Neuroticism | -0.02 | 678.30* | -1.60 | -22.50 |
|  | (-0.07, 0.04) | (93.50, 1263.20) | (-8.43, 5.22) | (-93.41, 48.42) |
| Neuroticism × Day | 0.00 | -1.47 | -0.00 | -0.02 |
|  | (-0.00, 0.00) | (-2.97, 0.02) | (-0.02, 0.02) | (-0.22, 0.17) |
| Openness | 0.01 | -274.20 | 1.81 | 0.70 |
|  | (-0.05, 0.06) | (-826.20, 277.80) | (-4.60, 8.22) | (-66.05, 67.44) |
| Openness × Day | -0.00 | -0.22 | -0.01 | -0.12 |
|  | (-0.00, 0.00) | (-1.68, 1.25) | (-0.03, 0.00) | (-0.32, 0.07) |
| Depression | -0.04 | -32.98 | 1.86 | -52.66 |
|  | (-0.13, 0.05) | (-934.70, 868.70) | (-8.56, 12.28) | (-160.90, 55.56) |
| Depression × Day | 0.00 | -0.14 | -0.01 | 0.02 |
|  | (-0.00, 0.00) | (-2.33, 2.06) | (-0.04, 0.02) | (-0.26, 0.30) |
| On campus (1 = yes) | 0.03* | 671.90*** | 5.38*** | 40.92** |
|  | (0.00, 0.06) | (408.80, 935.00) | (2.56, 8.20) | (11.47, 70.37) |
| Monday to Thursday (1 = yes) | 0.08*** | 966.20*** | 7.43*** | 73.67*** |
|  | (0.06, 0.09) | (817.30, 1115.00) | (5.80, 9.05) | (56.67, 90.67) |
| Friday or Saturday (1 = yes) | 0.19*** | 2042.30*** | 18.26*** | 265.10*** |
|  | (0.17, 0.20) | (1874.50, 2171.30) | (16.40, 19.63) | (245.80, 279.60) |
| Home football game day (1 = yes) | 0.30*** | 2865.20*** | 34.95*** | 498.80*** |
|  | (0.27, 0.33) | (2571.40, 3159.00) | (31.68, 38.21) | (464.5, 533.10) |
| Midterm break (1 = yes) | -0.08*** | -1711.30*** | -12.29*** | -95.48*** |
|  | (-0.10, -0.06) | (-1940.00, -1482.60) | (-14.75, -9.83) | (-121.30, -69.71) |
| Winter break (1 = yes) | -0.19*** | -3011.60*** | -22.64*** | -224.20*** |
|  | (-0.22, -0.16) | (-3352.70, -2670.60) | (-26.24, -19.03) | (-261.90, -186.50) |
| Summer break (1 = yes) | -0.02 | -1305.20*** | -9.41*** | -28.36 |
|  | (-0.05, 0.01) | (-1618.20, -992.20) | (-12.76, -6.07) | (-63.35, 6.63) |
| Thanksgiving holidays (1 = yes) | -0.22*** | -3681.00*** | -27.46*** | -247.50*** |
|  | (-0.26, -0.19) | (-4035.50, -3326.5) | (-31.34, -23.57) | (-288.20, -206.80) |
| Easter holidays (1 = yes) | -0.14*** | -2190.90*** | -13.14*** | -196.60*** |
|  | (-0.18, -0.10) | (-2614.20, -1767.70) | (-17.61, -8.67) | (-243.50, -149.70) |
| Orientation week (1 = yes) | 0.08*** | 397.30 | 2.38 | 192.70*** |
|  | (0.03, 0.12) | (-18.92, 813.40) | (-2.46, 7.22) | (145.30, 240.10) |
| Final exam week (1 = yes) | -0.06*** | -929.90*** | -7.31*** | -47.94*** |
|  | (-0.08, -0.04) | (-1173.50, -686.40) | (-10.00, -4.61) | (-76.20, -19.67) |
| Highest temperature (˚F) | 0.00*** | 23.68*** | 0.22*** | 2.87*** |
|  | (0.00, 0.00) | (17.87, 29.49) | (0.15, 0.28) | (2.20, 3.54) |

(*Continued*)

**Table 3.** (Continued)

| | Daily activity | Steps | Active minutes | Activity calories |
|---|---|---|---|---|
| Lowest temperature (°F) | -0.00* | -2.68 | -0.04 | -0.87* |
| | (-0.00, -0.00) | (-9.40, 4.04) | (-0.12, 0.03) | (-1.64, -0.09) |
| Precipitation in inch | -0.00 | -144.60* | -1.28 | -12.68 |
| | (-0.01, 0.01) | (-269.20, -19.94) | (-2.63, 0.07) | (-26.76, 1.39) |
| Snowfall in inch | 0.01* | 59.21* | 0.53 | 9.21** |
| | (0.00, 0.01) | (5.86, 112.60) | (-0.06, 1.13) | (2.99, 15.44) |
| Snow piling depth in inch | 0.00 | 16.55 | 0.45* | 2.65 |
| | (-0.00, 0.01) | (-17.82, 50.92) | (0.07, 0.83) | (-1.34, 6.64) |
| Sleeping minutes | 0.00*** | -5.36*** | 0.00 | 0.18*** |
| | (0.00, 0.00) | (-5.69, -5.02) | (-0.00, 0.01) | (0.14, 0.21) |
| Number of classes | 0.01*** | 278.80*** | 0.59** | 15.40*** |
| | (0.00, 0.01) | (238.60, 318.90) | (0.15, 1.03) | (10.78, 20.03) |
| Intercept | -0.22 | 15519.50*** | 36.14* | 768.30*** |
| | (-0.45, 0.01) | (13096.00, 17943.00) | (8.16, 64.12) | (476.90, 1059.80) |
| Number of cases | 38,821 | 39,334 | 43,580 | 43,671 |
| Number of individuals | 413 | 413 | 419 | 419 |
| Goodness-of-fit | | | | |
| AIC | 39232.34 | 768840.40 | 463399.00 | 669602.00 |
| BIC | 39720.64 | 769329.40 | 463893.90 | 670097.00 |
| Wald chi-square/df | 3552.40/52 | 8415.60/52 | 3384.67/52 | 5458.19/52 |
| Log-likelihood | -19559.17 | -384363.18 | -231642.50 | -334744.00 |

*$p < 0.05$

**$p < 0.01$

***$p < 0.001$ (two-tailed tests).

## Discussion

The study extends previous work on the factors that modulate PA among younger adults by conducting prospective analyses over a long study period (637 days) and by accessing effects of a variety of multilevel, multiscale factors on multiple PA measures at a fine temporal grain and obtained unobtrusively via wearables in the NetHealth project. Although previous studies unanimously report a downward trend in PA levels as young adults age [10–21], our analysis shows evidence of a declining pattern only for the daily steps. For all other PA measures, we find evidence of stable growth curves, net of other social, personal, psychological, environmental, and behavioral factors. Therefore, it seems like results reported in previous work may have overestimated downward temporal trends by failing to adjust for the proper set of correlates of PA across levels and time scales. For instance, when the initial BMI status and its effect on linear slope are *not* included in the models, we reproduce the familiar negative linear slope pattern for daily PA status and activity calories. This suggests that BMI status is an important personal factor that needs to be accounted for in future studies of the determinants of PA.

### Social influences on physical activity

A core contribution of the present study is a consideration of social factors in modulating PA levels, both social factors that are directed from others to the focal individual (peer-influence effects) and from the individual to others (social activity effects). We find, in line with previous studies [19, 26, 32–34], that there are peer influence effects on PA, with an individual's own

levels of PA increasing in tandem with the average PA levels of their daily contacts. Interestingly, we find that this peer-influence effect is stronger for male participants than for female participants in this sample, consistent with previous work showing that college-age men tend to spend more time together in the same physical locations and settings [59], and also consistent with previous research showing both differential sensitivity by gender of the effect of other people's physical activity on one own's physical activity, with men being much more likely to be influenced by other men [34]. The fact we are able to reproduce this result in the data speaks to the validity of the analysis, but also raises the issue as to why we observe this gender-based difference in peer influence effects on PA as one that should be investigated in future work.

Previous work suggests that the peer influence effect on PA may occur via face-to-face interaction that facilitates behavioral coordination, and leads to engagement in common activities. However, it is also possible that people can be influenced by the traits and behaviors of others without having to be exposed to those behaviors in a face-to-face setting (e.g., via communicative or normative pressures). To further explore this issue, we perform an additional analysis and find that only 9.3% pairs of peers show up at the same geolocation reported by their smartphones during the same time of a day across all cases. In other words, the peer influence on PA can operate without a face-to-face interaction or engaging in common activities. Two contacted participants don't have to do workout together to achieve a similar PA level. Moreover, because our PA measures of a participant's in-study contacts and the size of their social network can vary from day to day, these are strong findings, implying that daily changes in both network size and the PA levels of those in a participant's active network predict daily changes in his or her own PA patterns.

The fact that we find evidence of a "generalized friendship paradox" (GFP) [56–58] for PA in these data, may also explain the existence of indirect influence effects not mediated by direct face-to-face contact. According to GFP theory, the average levels of any dimensional trait among the people we are connected to (academic competence, beauty, and in our case PA) will always be higher than our own levels of the same trait. This means that, when individuals connect to others, they will definitionally be exposed to *descriptive normative standards* which suggest that their own levels of PA should be increased (because the average levels of contacts are perceived to be higher). This is a case in which the "majority illusion" produced by the GFP mechanism actually leads to a beneficial effect, namely, increased levels of PA on the part of the focal actor.

Finally, we find that another important social factor, namely, the overall level of social activity going from the person to others, as given by the number of daily contacts, is an important predictor of overall PA levels across all four indicators. "Social butterflies" are more physically active, while social isolates, or people with a more restricted set of outgoing ties are more sedentary.

## Personal, environmental, and behavioral factors

Of the personal factors predicting PA levels, female participants were found to be less active than male participants (on average), which corroborates previous literature [10, 13, 16, 17, 22, 33, 37]. With regard to ethnoracial status, findings are partly consistent with Suminski et al. [38]: while African American students were less active, Asian American students had about the same PA levels as white students. Importantly, our findings show that Hispanic students and students of other races had lower PA levels in this sample. Finally, consistent with prior research [39], our findings indicate that religious preference had minimal influence on PA levels. Overall, psychological factors were only moderately important in our analysis. While we

reproduce the long-standing finding that trait extraversion is linked to higher PA levels [41], we do not find a strong correlation between depression status and PA.

Turning to environmental factors, our models use on campus status as an indicator for both facilities access and direct peer influence, and it resulted in higher PA levels. Our findings also indicate that college students in the sample were least active on Sundays, somewhat active on Mondays to Thursdays, and most active on Fridays and Saturdays. It is possible that Behrens and Dinger [44] combine Saturday and Sunday together and thus lower the step counts on the weekend and conclude that college students walked more steps on weekdays. Most prior research only contains PA measures during school days. Our findings provide evidence that the NetHealth participants were a lot more active during home football game Saturdays, and much less active during Winter breaks and Thanksgiving among all breaks and holidays. As for weather indicators, we find that daily highest temperatures positively predicted PA levels. Some other suggestive findings include that on rainy days study participants walked fewer steps, but they were able to compensate for the loss in steps to reach the same level of daily PA status. And where the students spent about the same active minutes on snowing days, they walked more steps and burned more activity calories, which contributed to higher daily PA status on these days.

Finally, our findings indicate that when study participants got more sleep, they would walk fewer steps but burn more activity calories, and have higher daily PA status, though the magnitudes of these effects are very small. In addition, we find that when study participants took more classes that day, they walked more steps, spent more active minutes, burned more activity calories, and had higher daily PA status.

## Limitations and suggestions for future work

This study has a few limitations to note. First, the NetHealth project only collects data from one setting. Although most of our findings are consistent with prior literature, more longitudinal studies pertaining to US college students, especially those utilizing objective data, are needed to investigate how the findings herein would be different across samples. Second, this study simplifies the models by estimating the effects of local weather status on the college students' PA levels. Although the college students in this sample spent most of their days (i.e., 74%) on campus and we include in models an indicator about on-campus and off-campus statuses, theoretically local weather should have no effect on their PA levels when they were off campus. Future studies should use our findings pertaining to local weather effects with caution. Third, while most explanatory variables in our models are temporally stable (e.g. gender and race), relatively stable (e.g. psychological factors), or exogenous (e.g. environmental factors, number of classes taken daily), endogeneity bias might still arise when daily PA level affects sleeping minutes of a college student. Another set of latent growth-curve models will be estimated to address this reverse effect. But it is beyond the scope of this current study and we leave it for our future work. Finally, there was a significant amount of missing data as discussed in the sample attrition section. Not all participants had valid data every day and the total number of days of participation was not the same for all participants. Although missing data is a common issue in longitudinal study and the latent growth-curve models can take into account missing data during estimation, future inquiry is needed to understand how to increase the compliant levels of data collected by wearing devices.

## Study implications

Despite these limitations, our findings have notable implications. First, this study suggests both feasibility and merit in investigating the effects of multiple categories of factors, at several levels of analysis and temporal scales on PA statuses of college students over a long time period

in a unified modelling framework. Second, our work demonstrates the feasibility and productivity of moving beyond the self-report framework in the study of PA, while leveraging the use of unobtrusive, fine-grained data collection via wearables. As such, future work should aim at collecting valid, practical, and affordable objective data instead of less reliable self-report data [60]. Third, as the field moves to greater and better use of fine-grained unobtrusively collected data, our study demonstrates the advantage of linking such data to other traditional and archival data sources to produce multilevel, multiscale data capable of addressing multiple factors at once. We hope future work aimed at understanding temporal trends in individual PA and how its determinants extend the framework provided here beyond college-age young adults to more expansive samples including individuals at different stages in the life-course.

Our findings also have practical implications for PA intervention programs targeting college students. Above all, the preponderant role of BMI, both as an explanatory factor in our statistical models and in its influence in the effect estimate for temporal slopes suggest that this should be a prime target of interventions. Any program aimed at developing lifelong PA habits should take into consideration information on an individual's BMI, aiming to understand possible heterogeneity in response to program incentives based on this factor.

Moreover, our findings find strong evidence for the role social factors play in modulating PA levels. These factors operate in three ways: first, the influence goes directly from others to the individual, taking the form of a standard peer-influence effect; second, there is a "Generalized Friendship Paradox" effect, whereby the levels of PA people are exposed to when they look at their friends will always be on average higher than their own; finally, there is the fact that social connectivity, or the effort to maintain a large number of daily contacts, in itself increases PA. In tandem, these findings suggest that simple social interventions aimed at increasing PA levels should focus on both exposure (e.g., encouraging people to find out about their contacts physical routines and habits) and general encouragement to connectivity (e.g., providing incentives for individuals to avoid social isolation and expand their peer networks).

That said, it is important to keep in mind that peer influence is a double-edged sword: while youth with high PA levels can lead their contacts to becoming more active, those with lower PA levels can also help their contacts to be more sedentary. Therefore, intervention programs should flexibly build upon the notion of behavioral diffusion inside social networks, with the consideration of possible factors that might affect the behavioral diffusion process in different directions. For example, youth with high PA levels can disseminate pro-exercise norms and it is important to increase the magnitude of peer influence by changing the strength and transmission of their ties inside the social networks. On the contrary, amplifying peer influence among students with lower PA level might not be a good thing, and intervention programs should seek to reset the anti-exercise norms through multiple mechanisms, including verbal persuasion, encouragement, advice, expectation, criticism, and vicarious learning.

## Conclusion

In sum, this study contributes to the interdisciplinary study of the determinants of PA by using fine-grained data aggregated at the daily level. Our findings highlight the importance of peer networks and associated descriptive norm in affecting the dynamics of PA levels among college students. Moreover, we find evidence that BMI-related factors cancel out the downward trend in daily PA status and activity calories. In addition, higher PA levels are found to be common among students more embedded into the external world (e.g., large network size, high extroversion levels, and more courses taken). Overall, our findings suggest a complex picture of multiple categories of factors shaping the evolving process of PA and the need for

adopting objective measures, and provide insight into the direct and moderated pathways through which BMI influences PA levels.

## Supporting information

**S1 Table. Items for measuring the big five factors in personality trait ratings.**
(DOCX)

## Author Contributions

**Conceptualization:** Cheng Wang.

**Data curation:** Cheng Wang.

**Formal analysis:** Cheng Wang.

**Funding acquisition:** Omar Lizardo.

**Methodology:** Cheng Wang.

**Project administration:** David S. Hachen.

**Supervision:** Omar Lizardo, David S. Hachen.

**Visualization:** Cheng Wang.

**Writing – original draft:** Cheng Wang.

**Writing – review & editing:** Cheng Wang, Omar Lizardo, David S. Hachen.

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
