## [Decision Letter · Decision Letter 0]

24 Aug 2020

PONE-D-20-21952

Using Fitbit Data to Examine Factors that Affect Daily Activity Levels of College Students

PLOS ONE

Dear Dr. Wang,

Thank you for submitting your manuscript to PLOS ONE. After careful consideration, we feel that it has merit but does not fully meet PLOS ONE’s publication criteria as it currently stands. Therefore, we invite you to submit a revised version of the manuscript that addresses the points raised during the review process.

Two experts have reviewed the manuscript. While both reviewers were generally supportive of the manuscript, they both have concerns regarding the novelty of the report. Currently the Discussion section is mostly repetition of the retuls. Therefore, I strongly encourage the authors to consider the following points in addition to reviewers' comments:

Most importantly, extend the Discussion and talk about possible underlying mechanisms and possible implications of the results. The references are very old. For example there are only two 2020 citations (both self-citation), none from 2019, and only one from 2018 and 2017. Please give a more recent account of the literature in both the Introduction and Discussion. Please add figures to make the data more tangible, otherwise the data is completely vague, despite the reported statistics. Please add additional information to table captions/notes. For example what is reported for Extraversion in Table 2? or daily activity in Table 1? or how come daily on campus status for each individual each day is such a large number? what are the values in parentheses in Table 3?

We look forward to receiving your revised manuscript.

Kind regards,

Amir-Homayoun Javadi, PhD

Academic Editor

PLOS ONE

Journal Requirements:

Reviewers' comments:

Reviewer's Responses to Questions

**Comments to the Author**

1. Is the manuscript technically sound, and do the data support the conclusions?

Reviewer #1: Yes

Reviewer #2: Yes

2. Has the statistical analysis been performed appropriately and rigorously? 

Reviewer #1: Yes

Reviewer #2: Yes

3. Have the authors made all data underlying the findings in their manuscript fully available?

Reviewer #1: Yes

Reviewer #2: Yes

4. Is the manuscript presented in an intelligible fashion and written in standard English?

Reviewer #1: Yes

Reviewer #2: Yes

5. Review Comments to the Author

Reviewer #1: Review for PONE-D-20-21952

Using Fitbit Data to Examine Factors that Affect Daily Activity Levels of College Students

In this manuscript, authors aim to study different factors affecting daily physical activity with the hypothesis that the main predictor is the peer influence. The authors have used a latent growth-curve models to investigate trends in data.

The topic of study is interesting and the authors have collected a large data set over the span of 637 days from 619 participants for a total of 269,057 days. This indicates not all subjects participated for the total duration of study.

There are essential pieces of information that are missing from the manuscript such as:

The authors should specify the power of their study, and discuss the attrition, the range and the average number of days a subject participated in this study. If the total number of days of participation is not the same for all subjects, please discuss the bias that might have been introduced as a result and if any steps were taken to minimize this resulted bias.

Authors should further state the exact type(s) of Fitbit device(s) being used, how the data was extracted from Fitbit device (did they use third party apps for extracting data, …).

What were the inclusion and exclusion criteria?

What were the underlying health factors, age range, …

“International student” is not a race. Authors should just specify the racial percentages as a whole independent of the subject being an international student or not. If authors would like to see the possible effect of being an international student on physical activity, then the category would be international vs American or something to that effect.

The authors should also state how the subject wore the Fitbit device, dominant vs nondominant hand, continuously between charging or only during daytime, …

If more than one type of Fitbit devices were used, how similar was the measurement method of the extracted parameters between different devices. In other words, could the variation in Fitbit device have any effect on the measurement outcome and finding of this paper.

My understanding is that the Fitbit algorithm is ever evolving and as such the authors should discuss how the possible update in Fitbit firmware during the course of the study, if any, could have affected the analysis.

How was “Notre Dame friends” defined? It is not clear from the manuscript how Notre Dame friends’ average daily PA was measured. Were subjects recruited from friend groups?

It would be helpful to the readers, if possible, to include the list of psychometric items (The Big Five Inventory) in the paper or as a supplemental material. How were a standardized factor score for each item calculated?

Please specify how sleep minutes was determined.

Another element that is missing in this study is consideration of confounding factors. These factors could possibly include a shared interest (between the participants and their “Notre Dame friends”) or habits such as drug use or drinking, or their interest in nature hikes which all have the potential to play a role in participants and their “Notre Dame friends” level of physical activities. In other word, is the “Notre Dame friends” influence on the participants due to a true influence from the friends or is it due to similarity in their interests/behaviors such as interest in outdoorsy or night-time activities with then their influence on their level of physical activity. For example, could the participants and their “Notre Dame friends” spent the day studying together and have a less active day? In short please discuss the possibility of confounding factors.

Please include a brief description of Fitbit derived parameters.

Authors have analyzed data assuming a linear trend and also using a quadratic term. Based on what were these trends selected? Could a piecewise linear trend have been a suitable choice?

Please provide a visualization of data.

Authors have used the term time constant for one of the terms in equations 2&3. Time constant has a specific meaning and carries a unit of 1/unit of time. Are γ01 and γ11 in the unit of time? The terminology is not clear.

In Table 3, the number of cases and number of individuals are different for each of the four columns (daily activity, steps, active minutes, and activity calories) and much smaller than the totals from values provided in Tables 1&2. Please provide an explanation.

Minor comments:

In Abstract the authors probably meant to say under-studied and not understudied.

Authors have used the term with-subject. Did they mean to say within-subject?

Reviewer #2: Review for:

“Using Fitbit Data to Examine Factors that Affect Daily Activity Levels of College Students”

Cheng Wang, Omar Lizardo, David S. Hachen

In this paper, the authors present an observational analysis on the NetHealth project dataset to identify the effect of different factors on physical activity of college students. The authors use latent growth-curve model for estimation purposes. They show that physical activity is driven by peer effects and more general the students’ network. They also explore some heterogeneous effects with respect to the gender.

Despite the limitations of the current study in terms of generalization, the lack of novelty, and the identification of causality, the paper is well written, the study technically sounds, and the topic is interesting in the areas of social science, public health and human behavior. For these reasons, I think that the paper could be accepted for publication. I have some minor remarks:

- These latent growth models are linear in nature. And as the authors discuss in the manuscript, they assume a linear relationship between PA and temperature. Someone would argue that this is quite non-linear relationship. When the temperature is very high the PA could drop.

- I was expecting some discussion with respect to the endogeneity of the problem.

- It was not clear in the manuscript what is exactly the novel contribution here. Please extent on that. Also, it is not clear how this dataset was used otherwise from other scientists.

- I believe that Fig 1 is difficult to be followed by a non-expert of SEM. Given that PLoS ONE is an interdisciplinary journal, I was expecting more details/simplification.

- It would be more impactful (= more citations) if the authors release the replication code in github or somewhere similar (optional and in accordance to PLoS ONE recommendations/policy).

6. PLOS authors have the option to publish the peer review history of their article (what does this mean?). If published, this will include your full peer review and any attached files.

Reviewer #1: No

Reviewer #2: No

---

## [Author Response · Author response to Decision Letter 0]

8 Oct 2020

Response to Reviewers

PONE-D-20-21952 

Using Fitbit Data to Examine Factors that Affect Daily Activity Levels of College Students 

PLOS ONE

We would like to thank the editor and reviewers for their very constructive feedback. Incorporating the editor’s and reviewers’ suggestions for revision has resulted in a greatly improved manuscript. Below, we note the concerns of the editor and reviewers and then explain how we responded to each comment. 

Editor

1. Most importantly, extend the Discussion and talk about possible underlying mechanisms and possible implications of the results.

––––––––––––––––––––––––––––––––––––––––––––––––––––

As the editor suggested, we have now greatly expanded the discussion section which helps the reader get a sense of our findings, the possible underlying mechanisms, and theoretical and practical implications.

2. The references are very old. For example there are only two 2020 citations (both self-citation), none from 2019, and only one from 2018 and 2017. Please give a more recent account of the literature in both the Introduction and Discussion.

––––––––––––––––––––––––––––––––––––––––––––––––––––

We thank the editor for this suggestion. We now add 14 more citations after 2015 in the introduction and discussion, including Rebar et al. (2015), Roemmich et al. (2015), Friedenreich et al. (2016), Lerman et al. (2016), Li et al. (2016), Yang et al. (2016), Faust et al. (2017), Bopp et al. (2019), Cahuas et al. (2020), Contardo Ayala et al. (2019), Kraus et al. (2019), Elhakeem et al. (2020), Garritson et al. (2020), and Teychenne et al. (2020).

3. Please add figures to make the data more tangible, otherwise the data is completely vague, despite the reported statistics.

––––––––––––––––––––––––––––––––––––––––––––––––––––

As the editor and reviewer 1 suggested, we have now added a new Fig 2 which is a scatter and linear prediction plot with 95% confidence interval for each of the 4 dependent variables (i.e. physical activity, steps, active minutes, and activity calories). We discuss Fig 2 on page 12 in the revised manuscript.

4.1 Please add additional information to table captions/notes. For example what is reported for Extraversion in Table 2? 

––––––––––––––––––––––––––––––––––––––––––––––––––––

We have now added a new S1 Table including all the items for measuring the big five factors in personality trait ratings, including “Extraversion”.

4.2 What is reported for daily activity in Table 1? 

––––––––––––––––––––––––––––––––––––––––––––––––––––

Daily activity is a standardized factor score of 18 items, i.e., low range calories and minutes, fat burn calories and minutes, cardio calories and minutes, peak calories and minutes, steps, floors, sedentary minutes, lightly active minutes, fairly activity minutes, very active minutes, marginal calories, activity calories, calories BMR, and calories out. We have now added a note in Table 1 explaining how the daily activity indicator is constructed.

4.3 How come daily on campus status for each individual each day is such a large number? 

––––––––––––––––––––––––––––––––––––––––––––––––––––

Out of 269,057 daily cases, 199,981 (or 74.33%) took place when the participants were on campus. The editor is right that this number ends up being too large when it is reported as a variable for each individual each day. We now average the daily on campus status for each participant and report the new statistics (with a mean of 0.74 and standard deviation of 0.10) in Table 1.

4.4 What are the values in parentheses in Table 3?

––––––––––––––––––––––––––––––––––––––––––––––––––––

The values in parentheses in Table 3 are 95% confidence intervals. We have now changed the caption for Table 3 as “Parameters and 95% confidence intervals from the latent growth-curve models”.

Reviewer 1

1. The topic of study is interesting and the authors have collected a large data set over the span of 637 days from 619 participants for a total of 269,057 days. This indicates not all subjects participated for the total duration of study.

There are essential pieces of information that are missing from the manuscript such as:

The authors should specify the power of their study, and discuss the attrition, the range and the average number of days a subject participated in this study. If the total number of days of participation is not the same for all subjects, please discuss the bias that might have been introduced as a result and if any steps were taken to minimize this resulted bias. 

––––––––––––––––––––––––––––––––––––––––––––––––––––

This is a great question and set of suggestions, which are related to the comment 15. The reviewer is correct that not all participants had valid data every day and the total number of days of participation was not the same for all participants. We now add a “Sample attrition” section to discuss these on page 11 in the revised manuscript. 

First, the number of classes taken daily comes from the follow-up surveys conducted during winter 2016, summer 2016, winter 2017, and summer 2017, with response rates of 91%, 83%, 80%, and 76%, respectively. These unit non-response missing data lead to the difference in the number of individuals in the sample vs. in the models. 

Second, the difference in the number of cases can be explained by the threshold we set for the compliant rate, i.e., the number of minutes that a Fitbit device is on the wrist divided by total minutes of a day (1440). Following Faust et al. (2017) and Wang et al. (2020), we estimate the latent growth-curve models using data over the 637 days from participants who had relatively complete Fitbit data (defined as 80% or more daily within-person records) to guarantee the validity of PA measures (daily activity, steps, active minutes, and activity calories) and sleeping minutes. After applying the 80% threshold, over the 637 days we have data from the typical participant for 335 of those days, so about 47.4% of the daily Fitbit data is missing. 

Although missing data is a common issue in longitudinal study and the latent growth-curve models can take into account missing data during estimation, we now explicitly acknowledge that sample attrition is a limitation on page 19 in the revised manuscript.

2. Authors should further state the exact type(s) of Fitbit device(s) being used, how the data was extracted from Fitbit device (did they use third party apps for extracting data, …).

––––––––––––––––––––––––––––––––––––––––––––––––––––

Each NetHealth participant got a free Fitbit Charge HR wristband when joining the project. They agreed to install the Fitbit app on their smartphones and run it regularly to synchronize the physical activity and sleeping data to the Fitbit cloud, which were in turn backed up to a server administered by the NetHealth project team. We now introduce these on page 8 in the revised manuscript.

3. What were the inclusion and exclusion criteria?

––––––––––––––––––––––––––––––––––––––––––––––––––––

In fall 2015 the University of Notre Dame admitted 2007 full-time freshmen, among which 1069 were male students and 938 were female students. Based on their gender and racial composition, the NetHealth project team randomly selected 730 students and sent out invitations. Finally, 692 students accepted the invitations and completed the assent forms. For those under 18 years old their parents or guardians also signed the consent forms. Other than these, there were no particular inclusion or exclusion criteria.

4. What were the underlying health factors, age range, …

––––––––––––––––––––––––––––––––––––––––––––––––––––

In the current study we use the body mass index (BMI) as an indicator for a participant’s health status. The results from the entry survey indicated that only 10 (or 1.45%) participants said they had poor health status and 13 (or 1.88%) said they had a physical disability. Since those were rare events, we do not include them in the models. The ages of 99.6% participants range from 17 to 19, which were quite homogeneous.

5. “International student” is not a race. Authors should just specify the racial percentages as a whole independent of the subject being an international student or not. If authors would like to see the possible effect of being an international student on physical activity, then the category would be international vs American or something to that effect.

––––––––––––––––––––––––––––––––––––––––––––––––––––

We include “International student” due to the original sampling frame (available from https://www3.nd.edu/~instres/CDS/2015-2016/CDS_2015-2016.pdf) used by the NetHealth project team. As shown on page 3 of this file, nonresident alien was listed as one racial/ethnic category and didn’t overlap with other categories such as “Latino”, “Black or African American”, “White”, and “Asian American”. 

6. The authors should also state how the subject wore the Fitbit device, dominant vs nondominant hand, continuously between charging or only during daytime, …

If more than one type of Fitbit devices were used, how similar was the measurement method of the extracted parameters between different devices. In other words, could the variation in Fitbit device have any effect on the measurement outcome and finding of this paper. My understanding is that the Fitbit algorithm is ever evolving and as such the authors should discuss how the possible update in Fitbit firmware during the course of the study, if any, could have affected the analysis.

––––––––––––––––––––––––––––––––––––––––––––––––––––

Each NetHealth participant got a free Fitbit Charge HR wristband with a firmware version of 18.84 and a battery life of up to 5 days when joining the project. The participant made his or her own decision of wearing it on the dominant or nondominant hand and charging it daily, every other daily, or until the battery was drained. The NetHealth project team didn’t collect information about these statuses. During the 637 day there was only one firmware update, i.e., on March 8th of 2016 Fitbit released the firmware version 18.122 which provided minor bug fixes and stability improvement. This should not affect our analysis.

7. How was “Notre Dame friends” defined? It is not clear from the manuscript how Notre Dame friends’ average daily PA was measured. Were subjects recruited from friend groups?

––––––––––––––––––––––––––––––––––––––––––––––––––––

This is a critical comment. We have now changed “Notre Dame friends” to “in-study contacts” to avoid the confusion. These are other NetHealth participants where a focal participant interacts via smartphone each day. Since they are also on the project, their daily PA is reported by Fitbit devices as well.

8. It would be helpful to the readers, if possible, to include the list of psychometric items (The Big Five Inventory) in the paper or as a supplemental material. How were a standardized factor score for each item calculated?

––––––––––––––––––––––––––––––––––––––––––––––––––––

We thank the reviewer for making this suggestion. We now add the S1 Table including all the items for measuring the big five factors in personality trait ratings. 

We use the “alpha” command in Stata V15.2 to generate each standardized factor score from corresponding items. Taking “Extraversion” for example, first the sum of the 8 items for measuring extraversion are generated for each individual, then the variance of each item s_i^2 and that of the sum s_y^2 are calculated, and finally the Cronbach’s alpha is computed as α=(8/(8-1))((s_y^2-∑s_i^2)/(s_y^2)). If the Cronbach’s alpha is greater than 0.7, we get a scale with good internal reliability. To calculate each individual’ score on the scale, first the mean x ®_i and standard deviation s_i for each of the 8 items are generated, then each of the 8 items is standardized as (x_i-x ®_i)/(s_i), and finally the average of the 8 standardized items are computed as (∑(x_i-x ®_i)/(s_i))/8.

9. Please specify how sleep minutes was determined.

––––––––––––––––––––––––––––––––––––––––––––––––––––

A Fitbit device records minute-by-minute movement and heart rate data of its users. It uses a proven algorithm based on the movement pattern and heart rate variability (HRV) to determine the user’s sleeping status and reports the total sleeping minutes per day. We now introduce these on page 8 in the revised manuscript.

10. Another element that is missing in this study is consideration of confounding factors. These factors could possibly include a shared interest (between the participants and their “Notre Dame friends”) or habits such as drug use or drinking, or their interest in nature hikes which all have the potential to play a role in participants and their “Notre Dame friends” level of physical activities. In other word, is the “Notre Dame friends” influence on the participants due to a true influence from the friends or is it due to similarity in their interests/behaviors such as interest in outdoorsy or night-time activities with then their influence on their level of physical activity. For example, could the participants and their “Notre Dame friends” spent the day studying together and have a less active day? In short please discuss the possibility of confounding factors. 

––––––––––––––––––––––––––––––––––––––––––––––––––––

As the reviewer has indicated, peer influence on physical activity could occur via face-to-face interaction facilitating behavioral coordination, or engagement in common activities. To address these confounding factors, we perform a further investigation and find that only 9.3% pairs of peers show up at the same geolocation reported by their smartphones during the same time of a day across all cases. In other words, this peer influence on physical activity can operate without a face-to-face interaction or engaging in common activities. Two contacted participants don’t have to do workout together to achieve a similar level of physical activity. Moreover, because our measures of the physical activity of a participant’s in-study contacts and the size of their social network can vary from day to day, these are strong findings, implying that daily changes in both network size and the physical activity levels of those in a participant’s active network predict daily changes in his or her own physical activity patterns. We now discuss these on page 17 in the revised manuscript.

11. Please include a brief description of Fitbit derived parameters.

––––––––––––––––––––––––––––––––––––––––––––––––––––

A Fitbit device aggregates the minute-by-minute movement and heart rate data of its user into 18 measures on physical activity per day, including low range calories and minutes, fat burn calories and minutes, cardio calories and minutes, peak calories and minutes, steps, floors, sedentary minutes, lightly active minutes, fairly activity minutes, very active minutes, marginal calories, activity calories, calories BMR, and calories out. The Fitbit device also uses a proven algorithm based on the movement data and heart rate variability (HRV) to determine the sleeping status of its user and reports the total sleeping minutes that day. We now introduce these on pages 7-8 in the revised manuscript.

12. Authors have analyzed data assuming a linear trend and also using a quadratic term. Based on what were these trends selected? Could a piecewise linear trend have been a suitable choice?

––––––––––––––––––––––––––––––––––––––––––––––––––––

As repeated in the response to the next comment, we add a Fig 2 showing the scatter plot and linear prediction plot with 95% confidence interval for each of the four dependent variables. While there were more outliers on each of the four dependent variables during fall semester 2015 and fall semester 2016 due to the home football games and PA levels were generally low during two winter breaks and one summer break, we do not see an evident piecewise linear trend. We discuss this on page 12 in the revised manuscript.

Of course, we cannot rule out the possibility that the NetHealth participants can be categorized into subgroups with various piecewise linear trends. This issue can be addressed by applying growth mixture models with MPLUS. However, it is beyond the scope of the current study and we will leave it for our future work.

13. Please provide a visualization of data.

––––––––––––––––––––––––––––––––––––––––––––––––––––

As the reviewer has suggested, we now add a Fig 2 showing the scatter plot and linear prediction plot with 95% confidence interval for each of the four dependent variables in the revised manuscript.

14. Authors have used the term time constant for one of the terms in equations 2&3. Time constant has a specific meaning and carries a unit of 1/unit of time. Are γ01 and γ11 in the unit of time? The terminology is not clear?

––––––––––––––––––––––––––––––––––––––––––––––––––––

In equations (2) and (3), z is the time-constant variable (e.g. personal factors, cognitive factors) observed via surveys and staying consistent over the study period. η0 and η1 are latent variables that are not directly observed but estimated from the latent growth-curve model. γ01 and γ11 are not time-constant variables, but the magnitudes of linear effects that time-constant variable z has on the intercept term η0 and linear slope term η1.

15. In Table 3, the number of cases and number of individuals are different for each of the four columns (daily activity, steps, active minutes, and activity calories) and much smaller than the totals from values provided in Tables 1&2. Please provide an explanation.

––––––––––––––––––––––––––––––––––––––––––––––––––––

This is a great question and related to comment 1. First, the number of classes taken daily comes from the follow-up surveys conducted during winter 2016, summer 2016, winter 2017, and summer 2017, with response rates of 91%, 83%, 80%, and 76%, respectively. These unit non-response missing data lead to the difference in the number of individuals. Second, the difference in the number of cases can be explained by the threshold we set for the compliant rate, i.e., the number of minutes that a Fitbit device is on the wrist divided by total minutes of a day (1440). Following Faust et al. (2017) and Wang et al. (2020), we estimate the latent growth-curve models using data over the 637 days from participants who had relatively complete Fitbit data (defined as 80% or more daily within-person records) to guarantee the validity of PA measures (daily activity, steps, active minutes, and activity calories) and sleeping minutes. The 80% threshold applies to both focal participants and in-study contacts, making the number of cases and number of individuals are different for each of the columns in Table 3. We now add a “Sample attrition” section on page 11 to discuss these in the revised manuscript.

16. Minor comments:

In Abstract the authors probably meant to say under-studied and not understudied.

––––––––––––––––––––––––––––––––––––––––––––––––––––

We are thankful for a careful reading of the manuscript. We now change “understudied” to “under-studied” in the abstract.

17. Authors have used the term with-subject. Did they mean to say within-subject?

––––––––––––––––––––––––––––––––––––––––––––––––––––

We are thankful for a careful reading of the manuscript. We now fix these typos in the revised manuscript.

Reviewer 2

1. These latent growth models are linear in nature. And as the authors discuss in the manuscript, they assume a linear relationship between PA and temperature. Someone would argue that this is quite non-linear relationship. When the temperature is very high the PA could drop.

––––––––––––––––––––––––––––––––––––––––––––––––––––

We thank the reviewer for this suggestion. In ancillary models we add the square term of daily highest temperature, or that of daily lowest temperature, or that of both. We find that (1) these curvilinear effects themselves are not statistically significant, (2) they make their corresponding linear terms be statically insignificant, and (3) they do not help decrease the AIC and BIC values. Therefore, we exclude them from the models presented in the manuscript. We now clarify this on pages 14-15 in the revised manuscript.

2. I was expecting some discussion with respect to the endogeneity of the problem.

––––––––––––––––––––––––––––––––––––––––––––––––––––

We thank the reviewer for this comment. In our latent growth-curved models, the time-constant covariates are either ascribed status assigned at birth (e.g. gender and race) or relatively stable over time (e.g. cognitive factors), and thus are not driven by daily PA levels and their changes. Among the time-varying variables, environment factors are exogenous, and number of classes taken daily by college students are predetermined and unlikely to be affected by daily PA levels and their changes. The only scenario that endogeneity problem could arise is daily physical activity might affect sleeping minutes of a college student. We need another set of latent growth-curved models to address this reverse effect. But it is beyond the scope of this current study and we will leave it for future work. We now address this in the limitation section on pages 19-20 in the revised manuscript.

3.1 It was not clear in the manuscript what is exactly the novel contribution here. Please extent on that. Also, it is not clear how this dataset was used otherwise from other scientists.

––––––––––––––––––––––––––––––––––––––––––––––––––––

This is a crucial comment. While previous studies have greatly contributed to our understanding of PA among college students, there are manifest gaps in the literature. First, in the systematic review, Keating et al. (2005) find that no longitudinal study has been conducted to study the changes in PA among college students as well as how they are associated with corresponding determinants. Second, they do not take into consideration of behavioral factors, such as sleep duration and number of classes taken by a college student, both of which may impact the PA levels and their changes over time. Third, Keating et al. (2005) acknowledge that while steps assessed by pedometers and calories measured by accelerometers are good measurements of PA, they are hard to be standardized so that the PA levels can be compared across studies. Finally, all the aforementioned factors have not been examined in a single modeling framework to avoid the spurious effect among multiple determinants. These gaps in literature can be largely explained by the fact that there is a paucity of objective data tracing the PA of college students around the clock in natural settings over a long period of time. The current study builds upon extant literature and uses the Fitbit data from the NetHealth project to address these gaps. We now clarify these contributions on pages 6-7 in the revised manuscript.

3.2 Also, it is not clear how this dataset was used otherwise from other scientists.

––––––––––––––––––––––––––––––––––––––––––––––––––––

The dataset is publicly available at the NetHealth Project website (http://sites.nd.edu/nethealth/). More than 20 papers have been published on peer-reviewed journals since 2017 (see http://sites.nd.edu/nethealth/papers-2/), highlighting possible perspectives for using this dataset. 

4. I believe that Fig 1 is difficult to be followed by a non-expert of SEM. Given that PLoS ONE is an interdisciplinary journal, I was expecting more details/simplification.

––––––––––––––––––––––––––––––––––––––––––––––––––––

As the reviewer has suggested, we now add notes to provide more details for the parameters shown in Fig 1 on page 35.

5. It would be more impactful (= more citations) if the authors release the replication code in github or somewhere similar (optional and in accordance to PLoS ONE recommendations/policy). 

––––––––––––––––––––––––––––––––––––––––––––––––––––

Again, a valid point. The key scripts for estimating latent growth-curve models in Stata are now available at https://github.com/socnetfan/pa/, as indicated on page 10 in the revised manuscript.

References

Faust L, Purta R, Hachen DS, et al. Exploring compliance: observations from a large scale Fitbit study. In: SocialSens’17 Proceedings of the 2nd International Workshop on Social Sensing. New York: ACM; 2017. pp. 55–60.

Wang C, Lizardo O, Hachen DS. Using Fitbit Data to monitor the heart rate evolution patterns of college students. J Am Coll Health. 2020. doi: 10.1080/07448481.2020.1775610.

---

## [Decision Letter · Decision Letter 1]

28 Oct 2020

PONE-D-20-21952R1

Using Fitbit Data to Examine Factors that Affect Daily Activity Levels of College Students

PLOS ONE

Dear Dr. Wang,

Thank you for submitting your manuscript to PLOS ONE. After careful consideration, we feel that it has merit but does not fully meet PLOS ONE’s publication criteria as it currently stands. Therefore, we invite you to submit a revised version of the manuscript that addresses the points raised during the review process.

Please pay attention the few points mentioned by the reviewers. In particular please expand on reviewer two's comment on the novelty of the research. I expect a much more extended explanation of what the novel contribution of the study is. Reviewer one also mentioned a few important points.  

We look forward to receiving your revised manuscript.

Kind regards,

Amir-Homayoun Javadi, PhD

Academic Editor

PLOS ONE

Reviewers' comments:

Reviewer's Responses to Questions

**Comments to the Author**

1. If the authors have adequately addressed your comments raised in a previous round of review and you feel that this manuscript is now acceptable for publication, you may indicate that here to bypass the “Comments to the Author” section, enter your conflict of interest statement in the “Confidential to Editor” section, and submit your "Accept" recommendation.

Reviewer #1: (No Response)

Reviewer #2: All comments have been addressed

2. Is the manuscript technically sound, and do the data support the conclusions?

Reviewer #1: Yes

Reviewer #2: Yes

3. Has the statistical analysis been performed appropriately and rigorously? 

Reviewer #1: Yes

Reviewer #2: Yes

4. Have the authors made all data underlying the findings in their manuscript fully available?

Reviewer #1: Yes

Reviewer #2: Yes

5. Is the manuscript presented in an intelligible fashion and written in standard English?

Reviewer #1: Yes

Reviewer #2: Yes

6. Review Comments to the Author

Reviewer #1: The authors have addressed many issues that were brought up in the first round of reviews, but there are some remaining concerns that need to be addressed.

1. Please include the average and range (max and min) for the number of days of data included in the model for subjects participating in this study. This should be average, min and max number of days for participants after exclusion of days that did not meet the 80% threshold.

2. In response to reviewers portion of the document, the authors indicate: “… based on their gender and racial composition, the NetHealth project team randomly selected 730 students … “ It is not clear how this selection took place and what steps were taken to prevent sampling bias. But more importantly I do not understand why this information is not included in the body of the manuscript. What were reasons/criteria chosen to select that 730 out of 2007 freshmen enrolled that year? Did they aim for a specific percent of let's say Asian students? Did they aim for a specific percent of male vs female? Please include this information in the body of the manuscript.

3. In response to reviewers portion of the document, the authors state that there was no exclusion criteria but at the same time they indicate that students identifying themselves to be in poor health or having disabilities were excluded from the model. These exclusion criteria should be included in the body of the manuscript.

4. It would be appropriate to include in the body of the manuscript the age range of the subjects or their average age and its standard deviation.

5. Regarding considering international student as a classification under racial background/ethnicity, I do understand the limitation the authors had to work by; regardless, please find some other form of classification that could apply.

Reviewer #2: Review second round for PONE-D-20-21952

“Using Fitbit Data to Examine Factors that Affect Daily Activity Levels of College Students”

Cheng Wang, Omar Lizardo, David S. Hachen

In this paper, the authors present an observational analysis on the NetHealth project dataset to identify the effect of different factors on physical activity of college students. The authors use latent growth-curve model for estimation purposes. They show that physical activity is driven by peer effects and more general the students’ network. They also explore some heterogeneous effects with respect to the gender.

Despite some limitations of the current study the paper is well written, the study technically sounds, and the topic is interesting in the areas of social science and public health. In addition, the manuscript has improved substantially taking into account the comments from the editor and the two referees. For those reasons, I think that the paper can be accepted for publication.

I have one final remarks:

In my question about novelty and how this study can be differentiated from previous knowledge on the topic, the authors responded by citing a 2005 review paper. It sounds like an old reference to base the novelty of a 2020 contribution.

7. PLOS authors have the option to publish the peer review history of their article (what does this mean?). If published, this will include your full peer review and any attached files.

Reviewer #1: No

Reviewer #2: No

---

## [Author Response · Author response to Decision Letter 1]

14 Dec 2020

Response to Reviewers

PONE-D-20-21952R1 

Using Fitbit Data to Examine Factors that Affect Daily Activity Levels of College Students

PLOS ONE

We would like to thank the reviewers for their very constructive feedback. Incorporating the reviewers’ suggestions for revision has resulted in a greatly improved manuscript. Below, we note the concerns of the reviewers and then explain how we responded to each comment. 

Reviewer 1

1. Please include the average and range (max and min) for the number of days of data included in the model for subjects participating in this study. This should be average, min and max number of days for participants after exclusion of days that did not meet the 80% threshold.

––––––––––––––––––––––––––––––––––––––––––––––––––––

Thanks for the suggestion. After applying the 80% threshold, the number of days of Fitbit data ranges from 1 to 278, with a mean of 148 and standard deviation of 75. We now report these numbers on page 12 in the revised manuscript.

2.1 In response to reviewers portion of the document, the authors indicate: “… based on their gender and racial composition, the NetHealth project team randomly selected 730 students …” It is not clear how this selection took place and what steps were taken to prevent sampling bias. But more importantly I do not understand why this information is not included in the body of the manuscript. What were reasons/criteria chosen to select that 730 out of 2007 freshmen enrolled that year? 

––––––––––––––––––––––––––––––––––––––––––––––––––––

This is a great question. The NetHealth project was supported by the National Institute of Health (NIH). The project team made an estimation of the maximum number of Fitbit devices that could be distributed among the participants based on the NIH budget, and the answer was 730. We now include these details on pages 7-8 in the revised manuscript.

2.2 Did they aim for a specific percent of let's say Asian students? Did they aim for a specific percent of male vs female?

––––––––––––––––––––––––––––––––––––––––––––––––––––

Yes. The NetHealth project team adopted a stratified sampling strategy and aimed for a specific percentage of each gender-race strata based on the sampling frame. We now clarify this on page 8 in the revised manuscript.

2.3 Please include this information in the body of the manuscript.

––––––––––––––––––––––––––––––––––––––––––––––––––––

We now include the following information on pages 7 to 8 in the revised manuscript:

“In fall 2015 the University of Notre Dame admitted 2007 full-time freshmen, among which 1,069 (or 53%) were male students, 938 (or 47%) were female students, 1,352 (or 67%) were non-Hispanic white students, 219 (or 11%) were Hispanic students, 80 (or 4%) were non-Hispanic African-American students, 111 (or 6%) were non-Hispanic Asian-American students, 3 (or 0%) were non-Hispanic American Indian or Alaska Native students, 93 (or 5%) were non-Hispanic students with two or more races, 143 (or 7%) were students of other races, and 6 (or 0%) were students with race/ethnicity unknown (the detailed sampling frame is available from https://www3.nd.edu/~instres/CDS/2015-2016/CDS_2015-2016.pdf.) The NetHealth project team adopted a stratified sampling strategy and aimed for a specific percentage of each gender-race strata based on the sampling frame. The project team also made an estimation of the maximum number of Fitbit devices that could be distributed among the participants based on the NIH budget and sent out invitations to 730 students. Finally, 692 students accepted the invitations and completed the assent forms.”

3. In response to reviewers portion of the document, the authors state that there was no exclusion criteria but at the same time they indicate that students identifying themselves to be in poor health or having disabilities were excluded from the model. These exclusion criteria should be included in the body of the manuscript.

––––––––––––––––––––––––––––––––––––––––––––––––––––

We are sorry for the confusion resulting from our previous response to reviewers. The 10 (or 1.45%) students self-reporting poor health status and 13 (or 1.88%) students with physical disabilities were not excluded from our model. We meant to say we did not include “Health status” and “Physical disability” as explanatory variables in our model.

4. It would be appropriate to include in the body of the manuscript the age range of the subjects or their average age and its standard deviation.

––––––––––––––––––––––––––––––––––––––––––––––––––––

The ages of 99.6% participants ranged from 17 to 19 when they joined the project in 2015, with a mean of 18.4 and standard deviation of 1.8. We now report these numbers on page 10 in the revised manuscript.

5. Regarding considering international student as a classification under racial background/ethnicity, I do understand the limitation the authors had to work by; regardless, please find some other form of classification that could apply.

––––––––––––––––––––––––––––––––––––––––––––––––––––

We now classify “International student” as “Other races” in the revised manuscript.

Reviewer 2

In my question about novelty and how this study can be differentiated from previous knowledge on the topic, the authors responded by citing a 2005 review paper. It sounds like an old reference to base the novelty of a 2020 contribution.

––––––––––––––––––––––––––––––––––––––––––––––––––––

We thank the reviewer for this suggestion. We now add Barnett et al. (2014), Sallis et al. (2015), Yen & Li (2019), Scott (2020), and Rowlands (2020) to address the most previous knowledge on the topic as well as gaps in literature.

References

Barnett NP, Ott MQ, Rogers ML, Loxley M, Linkletter C, Clark MA. Peer associations for substance use and exercise in a college student social network. Health Psychol. 2014; 33: 1134–1142.

Rowlands AV. Measuring physical activity with body-worn accelerometers. In: Brusseau TA, Fairclough SJ, Lubans DR, editors. The Routledge Handbook of Youth Physical Activity. New York, NY: Routledge; 2020. pp. 330–346.

Sallis JF, Owen N, Fisher E. Ecological models of health behavior. In: Glanz K, Rimer BK, Viswanath K, editors. Health Behavior: Theory, Research, and Practice, 4th Edition. San Francisco, CA: Jossey-Bass; 2015. pp. 465–486.

Scott JJ. Pedometers for measuring physical activity in children and adolescents. In: Brusseau TA, Fairclough SJ, Lubans DR, editors. The Routledge Handbook of Youth Physical Activity. New York, NY: Routledge; 2020. pp. 315–329.

Yen HY, Li C. Determinants of physical activity: A path model based on an ecological model of active living. PLoS ONE. 2019; 14: e0220314.

---

## [Editor Report · Decision Letter 2]

16 Dec 2020

Using Fitbit Data to Examine Factors that Affect Daily Activity Levels of College Students

PONE-D-20-21952R2

Dear Dr. Wang,

We’re pleased to inform you that your manuscript has been judged scientifically suitable for publication and will be formally accepted for publication once it meets all outstanding technical requirements.

Kind regards,

Amir-Homayoun Javadi, PhD

Academic Editor

PLOS ONE
---

## [Editor Report · Acceptance letter]

21 Dec 2020

PONE-D-20-21952R2 

Using Fitbit Data to Examine Factors that Affect Daily Activity Levels of College Students 

Dear Dr. Wang:

I'm pleased to inform you that your manuscript has been deemed suitable for publication in PLOS ONE. Congratulations! Your manuscript is now with our production department. 

Kind regards, 

on behalf of

Dr. Amir-Homayoun Javadi 

Academic Editor

PLOS ONE